# Safety and Perioperative Outcomes of Laparoscopic vs. Open Hepatectomy of Central-Located Liver Lesions: A Multicenter, Propensity Score-Matched, Retrospective Cohort Study

**DOI:** 10.3390/jcm12062164

**Published:** 2023-03-10

**Authors:** Bao Jin, Muyi Yang, Yinhan Wang, Gang Xu, Yuxin Wang, Yuke Zhang, Yitong Liu, Xinming Niu, Xiao Liu, Xueshuai Wan, Huayu Yang, Xin Lu, Xinting Sang, Yilei Mao, Zhixian Hong, Shunda Du

**Affiliations:** 1Department of Liver Surgery, Peking Union Medical College (PUMC) Hospital, Chinese Academy of Medical Sciences & Peking Union Medical College, Beijing 100730, China; 2Department of Hepatobiliary Surgery, Fifth Medical Center of Chinese PLA General Hospital, Beijing 100039, China; 3Peking Union Medical College (PUMC), Chinese Academy of Medical Sciences & Peking Union Medical College, Beijing 100730, China; 4Department of Liver Surgery and Liver Transplantation Centre, West China Hospital, Sichuan University, Chengdu 610041, China

**Keywords:** laparoscopic hepatectomy, central-located liver lesions, safety, short-term outcomes, propensity score-matched analysis

## Abstract

Background: Short-term outcomes of laparoscopic hepatectomy of central-located liver lesions (LHCL) compared with traditional open hepatectomy of central-located liver lesions (OHCL) remain unclear. The aim of this study was to explore the safety and efficacy of LHCL. Methods: A retrospective analysis was performed on 262 patients who underwent hepatectomies involving resections of liver segment II, IV or VIII from January 2015 to June 2021 in two institutions. Patients in the LHCL group were matched in a 1:2 ratio to patients in the OHCL group. Results: After propensity score-matched (PSM) analysis, 61 patients remained in the LHCL group and 122 patients were in the OHCL group. What needs to be mentioned is that although not significant, patients in the OHCL group had increased lesion size (4.3 vs. 3.6 cm, *p* = 0.052), number (single/multiple, 84.8%/15.2% vs. 93.4%/6.6%, *p* = 0.097), and number of liver segments involved (one/two/three, 47.3%/42.0%/10.7% vs. 57.4%36.1%/10.7%, *p* = 0.393). To ensure surgical safety, fewer patients in the LHCL group underwent vascular exclusion than those in the OHCL group (*p* = 0.004). In addition, LHCL was associated with lower blood loss (*p* = 0.001) and transfusion requirement (*p* = 0.004). In terms of short-term outcomes, the LHCL group had significantly lower levels of peak ALT (*p* < 0.001), peak DBIL (*p* = 0.042), peak PT (*p* = 0.012), and higher levels of bottom ALB (*p* = 0.049). Moreover, the LHCL group demonstrated quicker postoperative recovery, which was represented by shorter time to first flatus, time to oral intake, time to drain off, and hospital stay (all *p* < 0.001). Importantly, the LHCL group had a significantly reduced occurrence of postoperative complications (*p* < 0.001) and similar R0 resection rates (*p =* 0.678) when compared to the OHCL group. Conclusion: LHCL is associated with increased safety and better perioperative outcomes and thus could be recommended for patients with central space-occupying liver lesions when appropriately selecting the surgical procedure according to the total tumor burden and carefully handled by experienced surgeons. From the experience of our center, LHCL could be performed to solitary lesion involving liver segment IV/V/VIII, <5 cm, with good safety and feasibility.

## 1. Introduction

Since the first small wedge and subsegmental laparoscopic liver resection in the early 1990s [1], laparoscopic liver resection surgery (LLR) has seen great progress in recent years, the indications of which cover both malignancies and benign diseases for minor, major, and combined minor–major liver resections [2].Multiple studies supported LLR with comparative perioperative and improved long-term outcomes in contrast to open liver resections (OLR), indicating its safety and feasibility [2,3,4].

LLR is predominantly recommended in wedge/minor non-anatomical resections and left lateral liver resections, while in high-volume centers, major laparoscopic hepatectomies are also regarded as standard clinical practice [5,6]. LHCL is only performed in centers with abundant surgical experience, because it is technically challenging with a high risk of perioperative bleeding, postoperative bile leakage, and bile duct stenosis and dysfunction [2,7]. However, there are also studies indicating that meso-hepatectomy could replace extended hepatectomy for lower parenchymal loss with fewer postoperative complications and mortality [8,9]. 

Previous studies on LHCL have mainly been case reports or case series, with only one single-center retrospective analysis [10,11,12,13,14,15]. Further studies are needed to assess the strengths and weaknesses of LHCL. Therefore, the aim of this study was to compare the safety and perioperative outcomes of LHCL compared to OHCL using PSM and data collected from two centers.

## 2. Materials and Methods

### 2.1. Patient Population

We retrospectively reviewed patients who underwent LHCL or OHCL due to any type of central space-occupying liver lesions from January 2015 to June 2021 in two institutions (Peking Union Medical College Hospital and the Fifth Medical Center of Chinese PLA General Hospital). We used the Couinaud liver segmentation method as the selection criteria, and patients with liver lesions located in Couinaud segments 4/5/8 were included in this study. All patients underwent liver function evaluation before surgery, of whom only patients with lesions categorized as Child–Pugh class A were allowed to undergo hepatectomy. Exclusion criteria included age ≤18 and incomplete prognostic data. The applicability of a laparoscopic approach was confirmed by a minimally invasive laparoscopic team in each hospital and was dependent upon the size and location of the tumors. In general, patients in the laparoscopic surgery group had lesions smaller than 5 cm in size, and most of them were single lesions. Patients with a history of upper abdominal surgery were usually excluded from both the laparoscopic and open groups. Of the selected patients, cases of LHCL were matched in a 1:2 ratio to OHCL controls. Written informed consent was obtained from all the patients. This study was approved by the ethical committee of our hospital (No. S-K1837). The study was designed and carried out as per the guidelines laid down in the Declaration of Helsinki. This work has been reported in line with the STROCSS criteria [16].

### 2.2. Surgical Procedures

LHCL usually includes anatomical and non-anatomical resections. Anatomical resection refers to the complete removal of the left medial lobe and right anterior lobe of the liver, that is, segments IV, V, and VIII (S4, S5, S8) of the liver. Anatomical LHCL is difficult as it involves the separation of the liver parenchyma on the left and right sides, treatment of important duct structures in the deep liver parenchyma, as well as ensuring that the blood flow of the left lateral lobe and right posterior lobe is not affected. Non-anatomical LHCL refers to local or irregular resection of a lesion in the central area of the liver. However, generally, there is no need for anatomical resection of the hilum or complete resection of the S4, S5, and S8 segments. Therefore, the difficulty and risk of LHCL are within a controllable range. In this study, most patients underwent non-anatomical liver resection.

In LHCL, all patients underwent general anesthesia and were placed in a supine position with their lower limbs separated. Five ports were used. The observation port was located around the belly button, and the other four operation ports were fan-shaped and distributed around the liver lesions. Generally, two ports would be located in the upper right abdomen, and the other two ports would be located in the upper left abdomen. The pressure of carbon dioxide pneumoperitoneum is generally maintained around 12–14 mmHg. After pneumoperitoneum is established, laparoscopic exploration is first performed to rule out other organ diseases in the abdominal cavity. The central venous pressure is generally controlled below 5 cm H_2_O during surgery to reduce intraoperative venous bleeding.

For the non-anatomical LHCL procedure, the first step was to mobilize the liver appropriately. The hepatic round ligament, falciform ligament, and coronary ligament (if necessary) were then divided using a harmonic scalpel. A blocking zone was reserved at the first hepatic portal. The location of the lesion was then determined according to the preoperative imaging evaluation and the intraoperative ultrasound, following which the resection line was marked 1–2 cm from the edge of the lesion. The second step was the separation of liver parenchyma. The harmonic scalpel and cavitron ultrasonic surgical aspirator (CUSA) are most commonly used to isolate liver parenchymal tissues. Bipolar electrocoagulation and peptide clip clamping were used to treat smaller pipes on the sections, and hem-Lock clamping was used to treat larger pipes. When intraoperative bleeding was difficult to control, the first hepatic portal was blocked intermittently. After most of the liver parenchyma was transected, in some cases, the remaining part of the liver parenchyma was cut off with an Endo-GIA stapler. Finally, the liver sections were managed in order to prevent bleeding and bile leakage. We mostly used bipolar cautery and sutures, if necessary, to control bleeding. Suspected bile leakage was clipped or sutured.

Anatomical LHCL was mainly performed according to the procedures described by Machado et al. [14] and Zheng [17]. The most important step in this procedure was to dissect the Glisson pedicle of the left inner lobe and the right anterior lobe of the liver as well as to cut the liver parenchyma within the ischemic line to ensure the integrity of the blood flow of the remaining liver.

### 2.3. Variables

The demographic and clinicopathological characteristics investigated in this study were age, sex, height, weight, body mass index (BMI), comorbidities, type, number, location and size of occupying lesions as well as the pathological diagnosis. Variables assessing surgical safety and early outcomes were surgical duration, estimated blood loss, transfusion, duration of vascular exclusion, time to first ambulation, time to first oral intake, time to first flatus, time to drain removal, postoperative analgesia time and pattern, length of postoperative hospital stay, and post liver function evaluation compared to preoperative data. Perioperative morbidity was defined as any complication that occurred within 30 days of surgery or during the hospital stay. All perioperative complications were graded according to the Clavien–Dindo classification [18]. Perioperative mortality was defined as death by any causes occurring within 30 days postoperatively or during the hospital stay.

### 2.4. Statistical Analysis

#### 2.4.1. Statistical Software

Data management, statistical evaluation, and analysis were performed using R software (version 3.2.5, R Foundation for Statistical Computing, Vienna, Austria) and SPSS software (version 24.0; IBM, Armonk, NY, USA).

#### 2.4.2. Descriptive Statistics

Continuous variables were summarized as means ± standard deviation (SD) or median (interquartile range) after testing for normality using the One-Sample Kolmogorov–Smirnov Test, whereas categorical variables were reported as percentages. The Student’s t-test or the Mann–Whitney U test were used for comparisons of continuous variables as appropriate, whereas the χ^2^ test was used to compare categorical data. The data were compared before and after the matching. Statistical significance was set at *p* < 0.05.

#### 2.4.3. Propensity Score Matching

Potential biases caused by different distributions of covariables in LHCL and OHCL cases were controlled by propensity score matching (MatchIt package), which included 14 preoperative covariates: sex, age, BMI, comorbidities (yes vs. no) including hypertension, diabetes mellitus, heart diseases, cerebrovascular disease, lung diseases, cirrhosis, and jaundice; ASA grade; and type, number, and location (number of IV/V/VIII liver segments involved) of occupying lesions. 

Each patient in the LHCL group was matched with two OHCL controls (1:2 ratio), without replacement, to minimize conditional bias. Each LHCL case was matched to its nearest score neighbor from the OHCL group. Multiple caliper widths were also measured. A caliper width of 0.2 resulted in the best trade-off between homogeneity and retained sample size. Finally, the patients were compared with respect to propensity scores.

## 3. Results

### 3.1. Patients Characteristics and PSM

A total of 262 patients were included from January 2015 to June 2021, of which 64 (24.4%) underwent LHCL and 198 (75.6%) underwent OHCL. Table 1 shows the clinical characteristics of the study cohort before and after matching. The LHCL and OHCL groups demonstrate significant discrepancy in lesion characteristics, with smaller lesion size (*p* < 0.001), smaller lesion number (*p =* 0.021) and lower involvement of segments IV/V/VIII (*p* < 0.001) in LHCL group. The other characteristics including sex, age, BMI, comorbidities (hypertension, diabetes mellitus, heart diseases, cerebrovascular diseases, lung diseases, cirrhosis, and jaundice), preoperative liver function (AST, ALT, ALB, TBIL, DBIL, PT, HBsAg), ASA grade, lesion size, lesion number, lesion location, lesion type (malignancy/benignancy, HCC, ICC, HCC-ICC, MLC, other malignancies/LH, FNH, LC, inflammatory nodules, and other benign lesions), tumor thrombus (PV, HV, no), vascular invasion, neural invasion, MVI, and tumor differentiation were not significantly different between the two groups (all p > 0.05). After PSM, we generated a balanced cohort of 61 patients that underwent LHCL and 112 patients that underwent OHCL. It is still worth noting that although not significant, the OHCL group had increased lesion size (4.3 vs. 3.6 cm, *p =* 0.052), number (single/multiple, 84.8%/15.2% vs. 93.4%/6.6%, *p =* 0.097), and number of liver segments involved (one/two/three, 47.3%/42.0%/10.7% vs. 57.4%36.1%/10.7%, *p =* 0.393), which to some extent implies that surgeons tend to choose laparocopic hepatectomy to deal with smaller solitary lesions, and the results and conclusions of our study are only applicable to those patients fully meeting the indications of different surgical procedures.

### 3.2. Outcomes after PSM

Table 2 lists the operation-related parameters and postoperative outcomes of both groups after matching.

In terms of surgical safety, fewer patients in the LHCL group underwent vascular exclusion than in the OHCL group (42.6% vs. 65.2%, *p =* 0.004), but patients in the LHCL group had a longer duration of vascular exclusion (34.5 min vs. 19.0 min, *p* < 0.001) as well as lower blood loss (*p =* 0.001) and transfusion requirement (9.8% vs. 28.6%, *p =* 0.004). Insignificant results included duration of operation (195.0 vs. 176.0, *p =* 0.075) and transfusion volume of RBC, plasma, and platelet (all *p* > 0.05). 

In terms of perioperative outcomes, the LHCL group showed significantly lower levels of peak ALT (221.0 U/L vs. 410.0 U/L, *p* < 0.001), peak DBIL (9.0 μmol/L vs. 11.0 μmol/L, *p =* 0.042) and peak PT (13.4 s vs. 14.2 s, *p =* 0.012) as well as a higher level of bottom ALB (34.0 g/L vs. 33.0 g/L, *p =* 0.049). Moreover, time to first flatus (2.0 days vs. 3.0 days, *p* < 0.001), time to oral intake (P < 0.001), time to drain off (5.0 days vs. 6.0 days, *p* < 0.001), and hospital stay (6.0 days vs. 9.0 days, *p* < 0.001) were significantly shorter in the LHCL group. The time to ambulation and analgesia patterns were comparable between the two groups (both *p* > 0.05). Although insignificant, the LHCL group had a lower level of postoperative peak TBIL (25.7 μmol/L vs. 29.0 μmol/L, *p =* 0.191), shorter duration of analgesia (2.0 days vs. 3.0 days, *p =* 0.429), and a higher number of patients who underwent R0 resection (98.0% vs. 96.8%, *p =* 0.678). Other insignificant results included higher in-hospital costs (55.84 thousand yuan vs. 45.62 thousand yuan, *p =* 0.053) in the LHCL group.

Moreover, the LHCL group had a significantly reduced occurrence of postoperative complications compared to the OHCL group (*p <* 0.001), especially the lower rate of Clavien–Dindo Grade I–II (8.2% vs. 46.4%) perioperative complications. Apart from ascites (0 vs. 11.6%, *p =* 0.001) and liver failure (0 vs. 6.3%, *p =* 0.012) which were significantly lower in the LHCL group, the rate of bile leakage (3.3% vs. 4.5%, *p =* 0.701), hemorrhage (1.6% vs. 6.3%, *p =* 0.135), abdominal (1.6% vs. 5.4%, *p =* 0.203), and wound infection/dehiscence (0 vs. 2.7%, *p =* 0.104) were also all lower in the LHCL group, although insignificant. In addition, pneumonia (3.3% vs. 2.7%, *p =* 0.823) was not significantly increased in the LHCL group.

## 4. Discussion

LLR, with magnified vision and delicate, minimally invasive operation, has been shown to be associated with reduced pain, blood loss, transfusion, complications, and hospital stay when compared to OLR of the left lateral and partial right liver [2]. The major advantage of LLR is that it avoids large incisions and subsequent long-term consequences (discomfort/pain, keloids, hernias, and cosmesis) and complications [19]. Moreover, no evidence supported that LLR increased postoperative death when carefully handled [20]. LLR is mostly applied to minor liver resection, especially left lateral liver resection or wedge resection for solitary lesions located in segment 2–6 [5,6], while major hepatectomies, such as right or left hepatectomy, was recommended to be best performed by experienced surgeons due to varying levels of difficulty with the scoring systems proposed [21,22]. LHCL handling lesions in the central liver segments IV, V, and VIII has traditionally been considered contraindicated owing to the difficult exposure of critical anatomic structures, high risk of massive bleeding, and challenging laparoscopic control of inflow and outflow while preserving supply to the adjacent segments II, III, VI, and VII [12]. Although increasing evidence has emerged for the safety and feasibility of LHCL according to short-term and long-term outcomes [10,11,12,13,14,15], LHCL remains a controversial technique.

In addition, few studies have evaluated the performance of LHCL vs. OHCL in the treatment of patients with all types of central liver lesions. We systematically searched PubMed for all articles published before October 2021 and found only one case control study comparing LHCL to OHCL [13]. It included 348 patients who underwent hepatectomy for HCC in the central liver segment between January 2012 and October 2017. After PSM analysis, 32 patients remained in the LHCL group and 96 patients remained in the OHCL group. In both adjusted and non-adjusted models, patients in the LHCL and OHCL groups had similar overall and disease-free survival rates. After PSM, LHCL was associated with a shorter hospital stay and reduced postoperative morbidity. Although this result revealed the superiority of LHCL for both short- and long-term outcomes, the small sample size of the LHCL group after PSM analysis and data from a single institution still impede the extrapolation of its conclusion. The other seven publications [10,11,12,14,15,23,24] describing LHCL are all case reports or case series from a single institution, with a small sample size and lack of OHCL control group, as listed in Appendix A. 

To our knowledge, our study is the largest multicenter case/cohort-matched LHCL-OHCL series using PSM to date, which demonstrated the safety and feasibility of LHCL in selected patients. In this study, we observed that LHCL had increased safety and better perioperative outcomes, with patients in the LHCL group displaying better postoperative liver function, quicker recovery, and shorter hospital stays after PSM.

Due to the small sample size (n = 158) of patients diagnosed with HCC in this cohort, we included all types of central liver-occupying lesions in our study, benign or malignant. PSM in this study covered all categorical covariates that could influence the prognosis of patients, including sex, age, BMI, comorbidities, ASA grade, and the type, number, and location of occupying lesions. Continuous variables such as lesion size were not included in PSM, as larger variables indicate a greater possibility of intraoperative trauma and bleeding. Preoperative liver functions of ALT, AST, ALB, TBIL, DBIL, PT, and HBsAg demonstrated a massive impact on postoperative recovery. Moreover, the pathological diagnosis of tumor thrombus, vascular invasion, neural invasion, MVI, and differentiation were not included in PSM covariates, because they are factors restricted to tumors. Nonetheless, these factors are strongly related to the malignancy of tumors and thus have a strong influence on the early outcomes of patients. Table 1 lists the covariates in PSM and other major confounding factors, as mentioned above, that are shown to be balanced after matching. 

As indicated by intraoperative outcomes, LHCL exhibited distinguished safety, with lower vascular exclusion requirements as well as lower blood loss transfusion requirement. LHCL enables better protection of blood vessels, for better visualization and timely hemostasis utilizing a series of blood control techniques and advanced instruments (e.g., linear stapler and electric coagulator). The challenging laparoscopic technique and delicate operation may also explain the longer duration of operation and duration of vascular exclusion. In addition, learning curve may provide another explanation for this question. In the early years, most cases with central-located liver lesions underwent open hepatectomy, while in later years, there was a shift toward more laparoscopic liver resections, with improved surgical skills and familiarity with meso-hepatic anatomy, resulting in a significant reduction in intraoperative bleeding and transfusion rates. 

Patients in the LHCL group showed better postoperative liver function and quicker recovery, especially shorter hospital stays, which is consistent with previous publications [13]. Overall, the LH group exhibited a significantly lower rate of postoperative complications and a lower Clavien–Dindo classification. As the severity of postoperative complications has been adopted as an important measure of quality in surgical studies, in addition to the frequency of complications, the results also represent the major advantages of LHCL. Regarding specific complications, LHCL was associated with a significant decrease in ascites and liver failure and an insignificant reduction in bile leakage, hemorrhage, abdominal and wound infection as well as an insignificant increase in pneumonia. Reduction in liver failure and ascites is likely related to reduced incision size, better preservation of the abdominal wall, and lower wound infection rate [25].

Compared to OLR, the chance of long-term survival after LLR remains controversial [26,27]. There was initial concern that LLR would decrease the tumor-free surgical margin because of the lack of palpation, as tumor-free surgical margins affect disease-free survival after resection of HCC [25,28]. With the development of laparoscopy-related technologies, such as the application of ultrasound and indocyanine green fluorescence navigation during laparoscopic surgery, this problem can be greatly improved [15,29,30]. In this study, the rate of R0 resection in the LHCL group was comparable to that of the OHCL group, which may be due to the widespread application of intraoperative ultrasound in laparoscopic surgery.

Although insignificant, reduced pain and higher in-hospital costs were also found in the LHCL group; the latter was explained by higher operating room costs for LLR than OR. Xu et al. suggested that it seems more reasonable to separate operating room costs from ward hospitalization costs in future studies [31]. Consistent with previous findings [13], our study proved that LHCL had comparable safety and improved perioperative outcomes compared to OHCL.

There are several limitations to the present study. First, although we have constituted the largest LHCL cohort with data from multicenter analysis by PSM, a small sample size and absence of randomization may still limit the strength and validity of the outcomes. Second, although inclusion of all kinds of central space-occupying liver lesions increases the scope of general application of this research, an imbalance in the number of different lesion types and a relatively small proportion of some diseases may weaken the reliability of the conclusion of this study when applied to these diseases. Third, due to incomplete records for the surgical procedures, no further information was able to be provided on the actual types of liver resection that were undertaken. However, the propensity score matching has included all preoperative covariates that could potentially influence the selection of liver resection types, including but not limited to comorbidities, ASA grade, and especially type, number, and location of occupying lesions, and after matching, the two cohorts stayed balanced on all the details. Therefore, we could draw the same conclusion for patients with similar lesion characteristics and undergoing appropriate choice of surgery procedures as our study shows. Fourth, the incomplete pathological reports resulted in a lack of histopathological information of underlying degree of liver fibrosis and/or cirrhosis, which may be associated with the increased postoperative ascites in the OHCL compared to the LHCL group. Last, we have to acknowledge the lack of follow-up impairs the comparison of long-term survival outcomes between LHCL and OHCL. With the accumulation of central liver HCC cases, long-term results will be included in our future research.

## 5. Conclusions

In conclusion, this multi-institutional, propensity score-matched study suggests that LHCL can be recommended as a reasonable method for patients with central space-occupying liver lesions considering its safety and superiority in perioperative outcomes. With accumulating experience of LLR and innovation in flexibility of laparoscopic techniques, we believe that LLR will have wider applications in the future.

## Figures and Tables

**Table 1 jcm-12-02164-t001:** Clinical characteristics of the study cohort before and after matching.

	Whole Cohort (n = 262)			Matched Cohort (n = 173)		
	LH (n = 64)	OH (n = 198)	*p*	LH (n = 61)	OH (n = 112)	*p*
Sex (male/female)	45 (70.3)/19 (29.7)	137 (69.2)/61 (30.8)	0.866	43 (70.5)/18 (29.5)	77 (68.8)/35 (31.3)	0.812
Age	53.9 ± 10.0	55.7 ± 11.5	0.278$	54.2 ± 9.9	54.7 ± 10.7	0.760 $
BMI (Kg/m^2^)	25.1 ± 3.2	24.4 ± 3.1	0.126$	25.2 ± 3.2	24.6 ± 3.3	0.258 $
Comorbidities (Yes/No)						
Hypertension	15 (23.4)/49 (76.6)	61 (30.8)/137 (69.2)	0.259	13 (21.3)/48 (78.7)	32 (28.6)/80 (71.4)	0.298
diabetes mellitus	12 (18.8)/52 (81.3)	36 (18.2)/162 (81.8)	0.919	10 (16.4)/51 (83.6)	18 (16.1)/94 (83.9)	0.956
Heart diseases	4 (6.3)/60 (93.8)	23 (11.6)/175 (88.4)	0.220	4 (6.6)/57 (93.4)	8 (7.1)/104 (92.9)	0.884
Cerebrovascular diseases	2 (3.1)/62 (96.9)	6 (3.0)/192 (97.0)	0.970	2 (3.3)/59 (96.7)	3 (2.7)/109 (97.3)	0.823
Lung diseases	1 (1.6)/63 (98.4)	6 (3.0)/192 (97.0)	0.503	1 (1.6)/60 (98.4)	2 (1.8)/110 (98.2)	0.944
Cirrhosis	35 (54.7)/29 (45.3)	83 (41.9)/115 (58.1)	0.074	33 (54.1)/28 (45.9)	53 (47.3)/59 (52.7)	0.394
Preoperative liver functions						
ALT (U/L)	22.0 (17.0, 38.0)	26.0 (17.0, 39.0)	0.585 #	25.0 (17.0, 29.0)	26.0 (17.0, 29.0)	0.735 #
AST (U/L)	27.0 (18.5, 37.0)	28.0 (21.8, 43.3)	0.051 #	27.5 (19.0, 37.0)	26.5 (21.0, 40.8)	0.396 #
ALB (g/L)	41.0 (38.3, 43.8)	41.0 (38.0, 44.0)	0.531 #	41.0 (38.5, 44.0)	42.0 (39.0, 44.0)	0.271 #
TBIL (μmol/L)	12.6 (10.1, 17.2)	13.7 (10.2, 18.2)	0.660 #	12.7 (10.2, 17.4)	14.2 (10.2, 19.2)	0.590 #
DBIL (μmol/L)	4.6 (3.2, 5.9)	4.8 (3.5, 6.1)	0.290 #	4.5 (3.3, 5.8)	4.8 (3.5, 6.6)	0.331 #
PT (s)	11.8 (11.2, 12.3)	11.8 (11.2, 12.6)	0.794 #	11.8 (11.2, 12.3)	11.8 (11.2, 12.6)	0.665 #
HbsAg (+/−)	37 (58.7)/26 (41.3)	96 (51.1)/92 (48.9)	0.291	35 (58.3)/25 (41.7)	57 (53.3)/50 (46.7)	0.528
ASA grade			0.296			0.642
I	34 (53.1)	85 (42.9)		31 (50.8)	49 (43.8)	
II	26 (40.6)	92 (46.5)		26 (42.6)	56 (50.0)	
III	4 (6.3)	21 (10.6)		4 (6.6)	7 (6.3)	
Lesion characteristics						
Lesion size (cm)	3.7 (2.3, 5.5)	5.0 (3.3, 8.0)	<0.001 #	3.6 (2.2, 5.5)	4.3 (3.0, 6.4)	0.052 #
Lesion number (single/multiple)	60 (93.8)/4 (6.3)	162 (81.8) /36 (18.2)	0.021	57 (93.4) /4 (6.6)	95 (84.8) /17 (15.2)	0.097
Lesion location			<0.001			0.393
Involving one of segment IV/V/VIII	38 (59.4)	54 (27.3)		35 (57.4)	53 (47.3)	
Involving two of segment IV/V/VIII	22 (34.4)	67 (33.8)		22 (36.1)	47 (42.0)	
Involving three of segment IV/V/VIII	4 (6.3)	72 (38.9)		4 (6.6)	12 (10.7)	
Malignancy/benign lesions	52 (81.3)/12 (18.8)	167 (84.3)/31 (15.7)	0.561	50 (82.0)/11 (18.0)	94 (83.9)/18 (16.1)	0.741
HCC	41 (64.1)	117 (59.1)		39 (63.9)	67 (59.8)	
ICC	5 (7.8)	24 (12.1)		5 (8.2)	12 (10.7)	
HCC-ICC	1 (1.6)	5 (2.5)		1 (1.6)	3 (2.7)	
MLC	4 (6.3)	18 (9.1)		4 (6.6)	11 (9.7)	
Other malignancies	1 (1.6)	3 (1.5)		1 (1.6)	1 (0.9)	
LH	4 (6.3)	17 (8.6)		3 (4.9)	8 (7.1)	
FNH	1 (1.6)	2 (1.0)		1 (1.6)	2 (1.8)	
LC	1 (1.6)	5 (2.5)		1 (1.6)	2 (1.8)	
Inflammatory nodules	5 (7.8)	5 (2.5)		5 (8.2)	4 (3.6)	
Other benign lesions	1 (1.6)	2 (1.0)		1 (1.6)	2 (1.8)	
Tumor thrombus *			0.263			0.383
PV	1 (1.9)	8 (4.8)		1 (2.0)	3 (3.2)	
HV	0 (0.0)	3 (1.8)		0 (0.0)	2 (2.1)	
No	51 (98.1)	155 (93.4)		49 (98.0)	89 (94.7)	
Vascular invasion (Yes/No) *	4 (7.7)/48 (92.3)	23 (13.9)/143 (86.1)	0.239	4 (8.0)/46 (92.0)	9 (9.6)/85 (90.4)	0.752
Neural invasion (Yes/No) *	2 (3.8)/50 (96.2)	5 (3.0)/161 (97.0)	0.770	2 (4.0)/48 (96.0)	3 (3.2)/91 (96.8)	0.803
MVI (Yes/No) *	23 (44.2)/29 (55.8)	54 (32.5)/112 (67.5)	0.123	21 (42.0)/29 (58.0)	29 (30.9)/65 (69.1)	0.181
Differentiation *			0.727			0.767
Low	4 (7.8)	12 (7.5)		4 (8.2)	9 (10.0)	
Moderately-low	7 (13.7)	12 (7.5)		7 (14.3)	8 (8.9)	
Intermediate	36 (70.6)	118 (74.2)		34 (69.4)	61 (67.8)	
Moderately-high	1 (2.0)	6 (3.8)		1 (2.0)	4 (4.4)	
High	3 (5.9)	11 (6.9)		3 (6.1)	8 (8.9)	

$, Student’s *t*-test; #, Mann–Whitney U test. ALT, alanine aminotransferase; AST, aspartate aminotransferase; ALB, albumin; TBIL, total bilirubin; DBIL, direct bilirubin; PT, prothrombin time; HBsAg, hepatitis B surface antigen; HCC, hepatocellular carcinoma; ICC, intrahepatic cholangiocarcinoma; HCC-ICC, combined hepatocellular carcinoma and intrahepatic cholangiocarcinoma; MLC, metastatic liver cancer; LH, liver hemangioma; FNH, focal nodular hyperplasia; LC, liver cyst; PV, portal vein; HV, hepatic vein; MVI, microvascular invasion. * Tumor thrombus (PV/HV/no), vascular invasion, neural invasion, MVI, and differentiation are restricted to malignancies.

**Table 2 jcm-12-02164-t002:** Intraoperative performance and perioperative outcomes of the study cohort after matching.

	Matched Cohort (n = 173)		
	LH (n = 61)	OH (n = 112)	*p*
Intraoperative			
Duration of operation (min)	195.0 (152.5, 280.0)	176.0 (150.0, 230.0)	0.075
Vascular exclusion (Yes/No)	26 (42.6)/35 (57.4)	73 (65.2)/39 (34.8)	0.004
Duration of vascular exclusion (min)	34.5 (25.8, 50.3)	19.0 (12.0, 27.0)	<0.001
Blood loss (mL)	200.0 (50.0, 200.0)	200.0 (100.0, 400.0)	0.001
Transfusion (Yes/No)	6 (9.8)/55 (90.2)	32 (28.6)/80 (71.4)	0.004
Transfusion volume			
RBC (U)	n = 5, 2.0 (2.0, 4.0)	n = 27, 2.0 (2.0, 4.0)	0.763
Plasma (mL)	n = 4, 500 (250, 600)	n = 24, 500 (400, 800)	0.465
Platelet (U)	n = 1, 1	n = 3, 1 (1, 2)	1.000
Post-operative			
Peak ALT (U/L)	221.0 (122.5, 347.5)	410.0 (239.8, 669.8)	<0.001
Bottom ALB (g/L)	34.0 (31.5, 37.0)	33.0 (31.0, 35.0)	0.049
Peak TBIL (μmol/L)	25.7 (19.7, 32.7)	29.0 (21.2, 37.5)	0.191
Peak DBIL (μmol/L)	9.0 (7.0, 12.3)	11.0 (7.1, 16.5)	0.042
Peak PT (s)	13.4 (12.9, 14.3)	14.2 (13.0, 15.1)	0.012
Resection margin (R1/R0)	1 (2.0)/49 (98.0)	3 (3.2)/92 (96.8)	0.678
Ambulation (d)	2.0 (1.0, 2.0)	2.0 (1.0, 2.0)	0.267
First flatus (d)	2.0 (2.0, 3.0)	3.0 (2.0, 3.0)	<0.001
Oral intake (d)	3.0 (2.0, 3.0)	3.0 (3.0, 4.0)	<0.001
Drain off (d)	5.0 (3.0, 6.0)	6.0 (5.0, 8.0)	<0.001
Analgesia (d)	2.0 (1.0, 3.0)	3.0 (0.0, 4.0)	0.429
Analgesia pattern			0.158
No	10 (16.4)	35 (31.3)	
NSAIDs	18 (29.5)	28 (25.0)	
Opioids	1 (1.6)	4 (3.6)	
PCA	0 (0.0)	2 (1.8)	
PCA+NSAIDs	27 (44.3)	36 (32.1)	
NSAIDs + Opioids	5 (8.2)	7 (6.3)	
Hospital stay (d)	6.0 (5.0, 8.0)	9.0 (7.3, 11.8)	<0.001
In-hospital costs (thousand yuan)	55.84 (43.23, 65.53)	45.62 (32.05, 69.61)	0.053
Total complications (n)	7 (11.5)	53 (47.3)	<0.001
Liver failure	0 (0.0)	7 (6.3)	0.012
Bile leak	2 (3.3)	5 (4.5)	0.701
Ascites	0 (0.0)	13 (11.6)	0.001
Hemorrhage	1 (1.6)	7 (6.3)	0.135
Pulmonary infection	2 (3.3)	3 (2.7)	0.823
Abdominal infection	1 (1.6)	6 (5.4)	0.203
Wound infection/dehiscence	0 (0.0)	3 (2.7)	0.104
Others *	1 (1.6)	9 (8.0)	0.058
Clavien–Dindo classification			<0.001
No	55 (90.2)	59 (52.7)	
Grade I–II	5 (8.2)	52 (46.4)	
Grade III–IV	1 (1.6)	0 (0.0)	
Grade V	0 (0.0)	1 (0.9)	
Postoperative mortality	0(0)	0(0)	-

RBC, red blood cells; NSAIDs, nonsteroidal anti-inflammatory drugs; PCA, patient-controlled analgesia. * Other complications include atrial fibrillation, pleural effusion, and urinary tract infection.

## Data Availability

The data that support the findings of this study are available from the Department of Liver Surgery in Peking Union Medical College Hospital, but restrictions apply to the availability of these data, which were used under license for the current study, and so they are not publicly available. Data are however available from the authors upon reasonable request and with permission of the Department of Liver Surgery in Peking Union Medical College Hospital.

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
