# Peer review of "Safety and Perioperative Outcomes of Laparoscopic vs. Open Hepatectomy of Central-Located Liver Lesions: A Multicenter, Propensity Score-Matched, Retrospective Cohort Study"

_jcm, 2023, doi:10.3390/jcm12062164_

Round 1

Reviewer 1 Report

This paper shows the results of the perioperative outcomes of laparoscopic versus open mesohepatectomy for all liver lesions by a propensity score-matched study.

Major revisions:

A)    Comparing with literature, there is no a significant novelty, therefore it would need to try a new corner to present the study

B)    It misses the long term follow up for cancers lesions. 

Minor revisions: 

A)    Table 1 needs to be simplified: jaundice and the level of bilirubin are the same thing and it needs just one of them; AST, ALT don’t tell more information… but it would be more appropriate to indicate the MELD and Child score. 

B)    Regarding histology information: it is important to add some findings regarding the non lesional liver (fibrosis, cirrhosis, steatosis,…)

C)    Blood loos are the same value between the two group but different statistically. It needs to review this data. 

Author Response

We are so appreciated for your letter on our manuscript entitled “Safety and perioperative outcomes of laparoscopic vs. open hepatectomy of central-located liver lesions: A multicenter, propensity score-matched, retrospective cohort study” (ID: jcm-2191445) and we are also extremely grateful to your comments on our manuscript. We have carefully considered every comment, and made cautious revision accordingly. Based on reviewer’s suggestions, we have answered the questions in detail one by one.

Response to the reviewer’s comments

Reviewer 1:

Comment 1:

Comparing with literature, there is no a significant novelty, therefore it would need to try a new corner to present the study.

Response 1:

We highly appreciate your review of our manuscript. We acknowledge that the perioperative outcome of laparoscopic versus open hepatectomy has been well investigated in the past. However, our study focused on the lesions in the central-liver region. As we all know, the anatomical structure of the central liver is very complex, which makes the operation of this site extremely difficult. Therefore, there are very few reports on laparoscopy in the central liver region. As far as we know, there is only 1 published cohort study on laparoscopic versus open mesohepatectomy (Ref: Surg Endosc. 2019 Sep;33(9):2916-2926. doi: 10.1007/s00464-018-6593-2), while this single-center study included only 32 patients of laparoscopic mesohepatectomy after propensity score-matched (PSM) analysis, which to some extent limits the extrapolation of conclusion. To our knowledge, our study is the largest multicenter case/cohort-matched LHCL-OHCL series using PSM to date, which demonstrated the safety and feasibility of LHCL in selected patients.

Comment 2

It misses the long term follow up for cancers lesions. 

Response 2:

We appreciate your suggestion. We strongly agree that the long-term follow-up of patients with malignancy should be supplemented. However, number of patients with malignant lesions in the laparoscopic group limits the statistical reliability of the long-term outcome analysis. Therefore, we did not show these data in this manuscript. With the accumulation of central liver HCC cases, long-term results will be included in our future research.

Comment 3:

Table 1 needs to be simplified: jaundice and the level of bilirubin are the same thing and it needs just one of them; AST, ALT don’t tell more information… but it would be more appropriate to indicate the MELD and Child score. 

Response 3:

Many thanks to the reviewers. We removed the jaundice section in the returned manuscript and kept only the bilirubin levels. However, we think it is necessary to retain ALT and AST levels, as what we would like to present in Table 1 is that clinical characteristics especially liver functions and lesion features are comparable after propensity-score matching, and ALT and AST could reflect liver damage. Also, the liver function of most patients is in a compensated state, which is child-pugh A grade, and there is no significant difference in MELD score between the two cohorts, selecting more detailed indicators such as bilirubin and PT can help the PSM model achieve better statistical efficiency.

Comment 4:

Regarding histology information: it is important to add some findings regarding the non lesional liver (fibrosis, cirrhosis, steatosis,…)

Response 4:

Thank you very much for the reviewer's suggestion. We strongly agree with you that it is meaningful to add some histological information. However, the incomplete pathological reports of the liver background surrounding the lesions of some patients limit the presentation of these findings.

Comment 5:

Blood loos are the same value between the two group but different statistically. It needs to review this data. 

Response 5:

Thank you for your careful review. The median of value of blood loss of LHCL and OHCL group are both 200ml, while the interquartile range are respectively (50,200) and (100,400), which indicates significantly less blood loss in the LHCL group and this result matches with the less percentage of transfusion in LHCL. 

We have tried our best to improve the manuscript and made some changes in the manuscript. These changes will not influence the content and framework of the paper and all changes have been marked in red in the revised paper. Finally, we appreciate very much for your time in editing our manuscript and the referees for their valuable suggestions and comments. I am looking forward to hearing from your final decision when it is made.

Reviewer 2 Report

I thank you for the opportunity to review this manuscript. Idea and research question of the article a clear. However, several issues need to be adressed.

1. Please delete the first sentence of the introduction

2. Please explain in the methods section why LHCL or OHCL were chosen. Which criteria were applied?

3. Despite the authors mentioning equality of the two groups (LHCL and OHCL) due to propensity-score matching, I have major concerns as to the comparability of the two groups and thus the conclusions drawn by the study. The OHCL group had larger tumors, more segments were involved and multiple tumors were more common. I propose reviewing the matching mehtods and applying stricter criteria in order to actually achieve more comparable groups.

4. There may be some mistakes in the tables concerning important numbers. E.g. blood loss is 200 ml for both groups, however p = 0.001 ?

Overall, I think this study needs profound revision of methods, language and scientific conclusions.

Author Response

We are so appreciated for your letter on our manuscript entitled “Safety and perioperative outcomes of laparoscopic vs. open hepatectomy of central-located liver lesions: A multicenter, propensity score-matched, retrospective cohort study” (ID: jcm-2191445) and we are also extremely grateful to your comments on our manuscript. We have carefully considered every comment, and made cautious revision accordingly. Based on reviewer’s suggestions, we have answered the questions in detail one by one.

 Reviewer 2:

Comment 1:

Please delete the first sentence of the introduction

Response 1:

We are very grateful to the reviewer for his suggestion that we have removed the first sentence of the Introduction section in the reworked manuscript.

Comment 2:

Please explain in the methods section why LHCL or OHCL were chosen. Which criteria were applied?

Response 2:

Thank you very much for the reviewer's comments. We used the Couinaud liver segmentation method as the selection criteria, and patients with liver lesions located in Couinaud segments 4/5/8 were included in this study. We have followed your suggestion and added relevant content to the Methods section, as highlighted in red.

Comment 3:

Despite the authors mentioning equality of the two groups (LHCL and OHCL) due to propensity-score matching, I have major concerns as to the comparability of the two groups and thus the conclusions drawn by the study. The OHCL group had larger tumors, more segments were involved and multiple tumors were more common. I propose reviewing the matching mehtods and applying stricter criteria in order to actually achieve more comparable groups.

Response 3:

We really appreciate your careful review. Propensity score matching (PSM) is a quasi-experimental method to construct an artificial control group by matching each treated unit with a non-treated unit of similar characteristics. After matching, the two groups (LHCL and OHCL) in this study had no significant difference in tumor size, involved segments and number of tumors (all P values > 0.05) , so we believe that the two groups are comparable. We can indeed increase the standard of matching, but this will make the number of cases in the two groups drop sharply, limiting the extensibility of our conclusions. Therefore, we believe that the current matching is reasonable and appropriate.

Comment 4:

There may be some mistakes in the tables concerning important numbers. E.g. blood loss is 200 ml for both groups, however p = 0.001 ?

Response 4:

Thank you for your careful review. The median of value of blood loss of LHCL and OHCL group are both 200ml, while the interquartile range are respectively (50,200) and (100,400), which indicates significantly less blood loss in the LHCL group and this result matches with the less percentage of transfusion in LHCL. 

We have tried our best to improve the manuscript and made some changes in the manuscript. These changes will not influence the content and framework of the paper and all changes have been marked in red in the revised paper. Finally, we appreciate very much for your time in editing our manuscript and the referees for their valuable suggestions and comments. I am looking forward to hearing from your final decision when it is made.

Round 2

Reviewer 1 Report

I think that the information mentioned in comment 4 would be significant and worthy to be reported. 

Author Response

We are so appreciated for your letter on our manuscript entitled “Safety and perioperative outcomes of laparoscopic vs. open hepatectomy of central-located liver lesions: A multicenter, propensity score-matched, retrospective cohort study” (ID: jcm-2191445) and we are also extremely grateful to your comments on our manuscript. We have carefully considered every comment, and made cautious revision accordingly. Based on reviewer’s suggestions, we have answered the questions in detail one by one.

Response to the reviewer’s comments

Reviewer 1:

Comment 1:

Regarding histology information: it is important to add some findings regarding the non lesional liver (fibrosis, cirrhosis, steatosis,…)

Response 1:

Thank you for your careful review and king suggestion. We strongly agree with you that histology information of non-lesional liver helps to understand the pathogenesis of patients and should be incorporated in the analysis. However, we have to admit that some patients in the cohorts have got incomplete pathological reports which do not include information of the lesion background, therefore, it is pity that this data could not be complemented. We express our deep apologies and will pay more attention to our data integrity in the future research.

We have tried our best to improve the manuscript and made some changes in the manuscript. These changes will not influence the content and framework of the paper and all changes have been marked in red in the revised paper. Finally, we appreciate very much for your time in editing our manuscript and the referees for their valuable suggestions and comments. I am looking forward to hearing from your final decision when it is made.

Reviewer 2 Report

Thank you for revising your manuscript. Overall, the authors responded adequately to the comments. However, the answer on comment 2 is not sufficient. There has been a missunderstanding of my comment: I did not want to know why a mesohepatectomy was performed, but how the decision between a laparoscopic or an open surgery was made. Please explain this decision process in the methods section (Surgeons' preference? Patient-related factor?). Thank you. 

Author Response

We are so appreciated for your letter on our manuscript entitled “Safety and perioperative outcomes of laparoscopic vs. open hepatectomy of central-located liver lesions: A multicenter, propensity score-matched, retrospective cohort study” (ID: jcm-2191445) and we are also extremely grateful to your comments on our manuscript. We have carefully considered every comment, and made cautious revision accordingly. Based on reviewer’s suggestions, we have answered the questions in detail one by one.

Response to the reviewer’s comments

Reviewer 2:

Comment 1:

I did not want to know why a mesohepatectomy was performed, but how the decision between a laparoscopic or an open surgery was made. Please explain this decision process in the methods section

Response 1:

Sorry for the misunderstanding and thank you for your patient explanation and review. The applicability of a laparoscopic approach was confirmed by a minimally invasive laparoscopic team in each hospital and was dependent upon the size and location of the tumors. In general, patients in the laparoscopic surgery group had lesions smaller than 5 cm in size and most of them were single lesions. Patients with a history of upper abdominal surgery were usually excluded from both the laparoscopic and open groups. We have also added the decision process into the methods section, as shown in red in line 73-77.

We have tried our best to improve the manuscript and made some changes in the manuscript. These changes will not influence the content and framework of the paper and all changes have been marked in red in the revised paper. Finally, we appreciate very much for your time in editing our manuscript and the referees for their valuable suggestions and comments. I am looking forward to hearing from your final decision when it is made.